# The Influence of Noise Level on the Stress Response of Hospitalized Cats

**DOI:** 10.3390/vetsci11040173

**Published:** 2024-04-12

**Authors:** Marisa Girão, George Stilwell, Pedro Azevedo, L. Miguel Carreira

**Affiliations:** 1Department of Clinics—Surgery, Faculty of Veterinary Medicine, University of Lisbon (FMV-ULisboa), 1300 Lisbon, Portugal; marisa.girao.vet@gmail.com (M.G.); stilwell@fmv.ulisboa.pt (G.S.); 2Centre for Interdisciplinary Research in Animal Health (CIISA), University of Lisbon (FMV-ULisboa), 1300 Lisbon, Portugal; 3Associate Laboratory for Animal and Veterinary Sciences (AL4AnimalS), 1300 Lisbon, Portugal; 4Anjos of Assis Veterinary Medicine Centre (CMVAA), 2830 Barreiro, Portugal; pedro.almeida.azevedo@gmail.com; 5Faculty of American Laser Study Club—ALSC, Altamonte Springs, FL 32714, USA; 6Private Human Dentistry (PHD), 1200 Lisbon, Portugal

**Keywords:** cat, behavior, stress, noise, cortisol, respiratory rate

## Abstract

**Simple Summary:**

A study examined how noise affects cats undergoing surgery in hospitals. Cats exposed to louder noises showed higher stress levels. Monitoring their behavior and breathing helps assess stress. Minimizing hospital noise is crucial for cat welfare, supporting guidelines to make their stay more comfortable. This study emphasizes the importance of minimizing noise levels in hospitals to improve the welfare of hospitalized cats, aligning with existing recommendations for enhancing feline well-being.

**Abstract:**

The study aimed to investigate the impact of noise levels in the hospital environment on the stress experienced by hospitalized cats undergoing elective ovariohysterectomy surgery. A total of 33 domestic female cats were included in the study, divided into four groups: a control group (CG) and three experimental groups based on ward noise levels: G1 (Quiet, <60 dB), G2 (Medium, between 60 and 85 dB), and G3 (Noisy, >85 dB). Behavioral assessments, respiratory rate (RR), and plasma cortisol levels ([Cort]p) were measured as indicators of stress. A composite measure of stress, termed the final stress value (FSV), was calculated by summing scores across various behavioral categories. Data collection occurred at three time points: immediately following surgery (T1), and at 2 h (T2) and 3 h (T3) post-surgery. FSV and RR were assessed at all three time points (T1, T2, and T3), while [Cort]p levels were measured at T1 and T3. The study observed that the median values of FSV, RR, and [Cort]p tended to increase with both higher noise levels and longer exposure durations to noise. Significant differences in RR were found between group pairs G1G2 at T1 (|D| = 0.63 < cut-off = 0.98), and G2G3 at T2 (|D| = 0.69 < cut-off = 0.97). Regarding [Cort]p, significant differences were noted between the CGG1 group pair at T0 (*p* < 0.01), and T3 (*p* = 0.03). Furthermore, an excellent Spearman correlation coefficient (rho = 0.91) was found between FSV and RR, indicating that RR can serve as an effective tool for assessing stress levels in cats. The findings of this study provide valuable insights into the welfare of cats in a hospital environment and support the scientific validity of existing recommendations aimed at improving their well-being. Specifically, the study underscores the importance of minimizing noise levels in hospitals as a means to reduce stress in cats. This conclusion aligns with existing guidelines and recommendations for enhancing the welfare of hospitalized cats.

## 1. Introduction

Over the last decade, the popularity of cats as pets has undergone a favorable evolution [1]. However, some of these animals do not receive regular veterinary care due to concerns about their reactive and intense behavior in response to stress at veterinary clinics [2,3]. Stress triggers agonistic behavior, commonly divided into offensive attack (fight) and defensive responses, the latter with variants of escape (flight) and behavioral inhibition (freezing) [4]. In cats, the stress response system (SRS) is an evolutionary and highly adapted mechanism that demonstrates efficiency in situations conveying threat, whether real or potential, by increasing natural defense capabilities through controlled physiological and behavioral changes [5,6,7]. However, the SRS can also contribute to the development of various physical disorders, besides inducing negative emotions, increasing the likelihood of psychological challenges [8]. Situations conditioned by negative emotions create intense and long-lasting contextual memories; hence, the initial experience in a given environment will strongly influence current and subsequent reactions [5,9]. The behavioral and physiological changes resulting from SRS activation make it possible to quantify the magnitude of stress in the organism. Feline individuality and a wide variety of stress sources are associated with a high range of behavioral quantification methods available in the literature. The use of an ethogram with a final stress value (FSV) associated, respiratory rate (RR), and plasmatic cortisol levels ([Cort]p) are potentially useful tools for assessing stress levels in cats [6,10]. Hospitalization is an inevitable experience in certain circumstances for a cat. In order to minimize adverse perceptions, the confinement surroundings and features must be adapted to the peculiarities of the species, as should the macroenvironment [11,12,13]. Recognizing potential environmental stress sources enables the adoption of simple but effective measures that benefit the animal by preventing the deleterious effects of stress and providing a lower probability of adverse outcomes from containment. This also benefits the relationship between the employees of the veterinary medical care center and the owners of that center, improving the productivity and efficiency of the service [14,15,16]. However, current recommendations for the feline species are largely based on empirical experience and are, therefore, considered as only simple indications [9,11,16,17]. Referenced in the literature on hospitalized cats, the study aims to assess the effect of a macroenvironmental parameter—noise level—in reducing the stress experienced by animals of that species.

## 2. Materials and Methods

A sample of 33 female domestic cats (*N* = 33), with an average age of 0.9 ± 0.1 years (ranging from 6 months to 2.7 years) and an average body weight of 3.5 ± 0.08 kg (ranging from 2.8 kg to 4.5 kg), underwent elective ovariohysterectomy (OVH) using the classic surgical technique. The study protocol was approved by the ethical review committee of the Faculty of Veterinary Medicine, University of Lisbon (CEIE, approval n° 001/2022/P1, 17 January 2022), and complies with the ARRIVE guidelines. Written informed consent was obtained from the institution responsible for the animals. Patients were admitted to the study after tutors signed the informed consent. All animals were housed in the same surgical ward at the Veterinary Teaching Hospital of the Faculty of Veterinary Medicine—University of Lisbon, where the study took place, before surgery. They were then randomly assigned to four groups: a control group (CG) and three additional groups (G1, G2, and G3). Behavioral assessments were conducted on cats in the ward cage, evaluating avoidance of not-to-contact behavior, body posture, and facial expressions at 5 and 10 min after entering the cage. Cats showing extreme nervousness or calmness were excluded. After surgery, the CG was placed in a room with noise levels < 50 dB, while the other groups (G1, G2, G3) were allocated to wards with noise levels categorized as Quiet (<60 dB), Medium (60–85 dB), and Noisy (>85 dB). All the cats were exposed to multiple environmental stimuli in the hospital environment within the wards, reflecting real-world conditions in a veterinary setting hospital. Environmental noise measurements were taken using a Sonometer S01^®^ model from GESA Termómetros (Urduliz, Spain). All patients received the same pre-medication with non-steroidal anti-inflammatory drugs (meloxicam at 0.2 mg/kg; SC) and opioids (buprenorphine at 0.02 mg/kg; IM), and anesthesia induction with ketamine (5 mg/kg; IM) used in combination with dexmedetomidine (40 µg/kg; IM) followed by anesthesia maintenance with isoflurane with a dose of 1.0%. Atipamezole was administered to reverse dexmedetomidine effects post-surgery. The surgical procedure was always performed by the same surgeon to reduce BIAS, with a mean duration of 12 ± 5.7 min. Blood samples were taken from the cephalic vein after topical application of bupivacaine ointment on the skin, and a volume of 1 mL was collected at T1 and T3. Cats were not anesthetized for blood sample collection at T3, and their handling was easily performed by the veterinary surgeon and nurse. Behavioral assessments and respiratory rate (RR) were evaluated at three time points: immediately after surgery (T1), and 2 (T2) and 3 h (T3) post-surgery; and the plasmatic cortisol level ([Cort]p) was measured at T1 and T3. A simplified ethogram was used for behavioral assessment based on the Cat Stress Score (CSS) (Table 1), and it was necessary to analyze repeatability to validate it. Thus, a total of six observers analyzed the behavior of an independent cat undergoing the same surgical and therapeutic protocol at T2. The final stress value (FSV), calculated from behavioral scores, and RR were assessed at all three time points considered, namely T1 (1 h), T2 (2 h), and T3 (3 h) after the surgery ended. The behavioral information was registered after a 5 min period spent in front of the hospital cage, attempting to mitigate the influence of the observer’s presence. The recording was performed at the three considered time-points, T1, T2, and T3, after the end of surgery, always by the same person, a veterinarian surgeon, in order to reduce BIAS. The respiratory rate was obtained by visualizing and counting the number of thorax and/or abdominal movements during a period of 1 min, and was assessed at T1, T2, and T3. Blood samples used to measure plasma cortisol ([Cort]p) levels were collected in EDTA tubes and centrifuged at 5000 rotations per minute. The resulting plasma was transferred to Eppendorf tubes and stored at −20 °C. Subsequently, all plasma samples were analyzed using a commercial ELISA kit called DRG^®^ EIA-1887 system (DRG Instruments GmbH, Marburg, Germany), previously validated for cats. Statistical analyses were conducted using the R^®^ program, with significance set at *p* < 0.05. Ethogram validation included intraclass correlation coefficient (ICC) analysis. The agreement between cortisol measurement methods was assessed using the Bland–Altman method, and group comparisons were made using Kruskal–Wallis and ANOVA tests. The correlation between stress markers was evaluated using the Spearman coefficient.

## 3. Results

Data from the total sample (CG, G1, G2, and G3) showed mean and standard deviation values for each ethogram category: Activity (1.17 ± 0.17), Body (1.0 ± 0.0), Eyes (1.0 ± 0.0), Pupils (1.0 ± 0.0), Vibrissae (0.0 ± 0.0), Ears (0.83 ± 0.17), Tail (1.5 ± 0.0), and Final Stress Value (FSV) (6.5 ± 0.26). Repeatability analysis using the intraclass correlation coefficient (ICC) resulted in a high value (0.98), indicating strong agreement. The Shapiro–Wilk test confirmed normal distribution for both techniques used to measure [Cort]p, with mean values of 45.15 ± 7.78 ng/mL for the ELISA kit EIA-1887 and 35.0 ± 7.63 ng/mL for the chemiluminescent ELISA system (Table 2). The Bland–Altman plot showed a bias of −7.2 between the techniques, with no statistical significance (*p* = 0.1) (Figure 1). However, the limits of agreement were wide (−24.5 to +10.2). The FSV, RR, and [Cort]p at T1, T2, and T3 did not follow a normal distribution (*p* > 0.05). Median values of FSV, RR, and [Cort]p tended to increase with noise level and exposure duration. Kruskal–Wallis tests revealed significant differences between groups for FSV and RR at all time points (*p* < 0.01). ANOVA One-Way tests showed increased [Cort]p means across all groups, with significant differences at T1 (*p* < 0.01) and T3 (*p* = 0.03) (Table 3). Post-hoc DSCF tests identified significant differences in RR values between groups G1 and G2 at T1, and between G2 and G3 at T2. Additionally, the Bonferroni test found differences between CG and G1 at T1, and CG and G1 at T3. No significant differences were found between time points for any group. Spearman correlation showed poor correlation between [Cort]p and FSV (rho = 0.09) and [Cort]p and RR (rho = 0.20), but excellent correlation between FSV and RR (rho = 0.91) (Table 4).

## 4. Discussion

The study was conducted on a sample of female domestic cats undergoing elective ovariohysterectomy surgery, with no complications or idiosyncrasies recorded during the post-operative period. In this study, four groups of cats were examined: CG cats were housed in a room isolated from other animals, while G1, G2, and G3 cats were accommodated in the hospital ward with varying environmental noise levels. Before the experiment, all animals experienced stressful events, including food deprivation, containment in carriers, transportation to a veterinary hospital, disruptive sounds in the surgical preparation room, drug administration, anesthesia, and surgical techniques. The recorded sound intensity was spontaneous and, therefore, has the advantage of reflecting the genuine ward environment in a hospital; however, it was not standardized for all hospitalized individuals. The ethogram used for behavior assessment was adapted from the Cat Stress Score [18,19,20], with inter-observer reliability achieved through preliminary tests (ICC). The strong correlation between observers’ records in each ethogram category ensured excellent agreement between individuals. Combined with the scale’s repeatability, strict confidence limits, and low average variance, the ethogram used was validated, reducing the potential error associated with each animal’s individuality under evaluation [10,21]. During hospitalization, most CG cats were considerably more relaxed than those in G1, G2, and G3. CG cats had significantly lower FSV medians than G1 cats at T3 and lower than G2 and G3 cats at T1, T2, and T3. Consequently, they interacted easily with humans and exhibited docile behavior, requiring less laborious manipulation, often only necessitating one person to collect blood samples and remove the venous catheter. Before surgery, all animals experienced similarly stressful events, and after surgery, the ward’s macroenvironment remained consistent except for the noise level. The Bland–Altman plot revealed a consistent non-zero bias, indicating that, on average, the ELISA EIA-1887 kit measured 7.2 ng/mL higher than the laboratory method. Adjusting for this bias, it was possible to normalize the kit measurements by subtracting 7.2 units [22]. However, after applying the Bland–Altman method in conjunction with the paired *t*-test, the difference between the techniques was not statistically significant, suggesting that the bias was effectively minimized. However, the limits of agreement were considerably wide and clinically unacceptable, suggesting potential overestimation of values by up to 24.5 ng/mL. This discrepancy may primarily stem from the small sample size. Cortisol levels exhibited a consistent pattern across all groups (CG, G1, G2, and G3), increasing from T1 to T3, indicating activation of the hypothalamic–pituitary–adrenal (HPA) axis. Throughout the post-operative period, the HPA axis continued to experience sustained stimulation, irrespective of the group, due to the cats’ exposure to an unfamiliar environment. However, no statistically significant differences between time points were observed in these variations. This lack of significance could be attributed to two factors: (1) the potential interference of the ward’s macroenvironment, and (2) an ineffective hormonal response to the various intensities of stress. Armario et al., (2012) [23] explained that cortisol levels may reach a plateau after intense activation due to factors such as the depletion of adrenocortical esters stored at cytoplasmic lipid droplets or the absence of enzyme precursors necessary for additional glucocorticoid synthesis. These findings align with those of Stella, Croney, and Buffington (2013) [11], who similarly concluded that there are no significant changes in plasma cortisol levels after repeated exposure of cats to stress factors. However, significant statistical differences between the CG and G1 groups were observed at both T1 and T3 when comparing cortisol levels ([Cort]p). According to Schulte (2014) [24], environmental disturbances do not seem to induce a mediated cortisol response in fish, which may be attributed to hypothalamic–pituitary–adrenal (HPA) axis activation. It is widely acknowledged that physical stressors can lead to habituation of the cortisol response over time, meaning that repeated exposure does not necessarily result in an increased cortisol response [25,26,27,28]. Some cytoprotective drugs, particularly proton pump inhibitors, have the potential to either increase or suppress HPA axis activity. Since none of these drugs were administered in the study, cortisol release and its plasma levels were not affected [29,30,31]. A trend toward increased final stress value (FSV) and respiratory rate (RR) values from T1 to T3 was observed for groups G1, G2, and G3, with no statistically significant differences between the pairs of groups G1G3 and G2G3 at T1. This suggests a similar level of stress across these groups. In contrast, the group pair G1G2 exhibited a different pattern. G1 showed a lower mean final stress value (FSV), indicating a tendency towards freezing behavior, where the cat appears relaxed but physiological parameters such as respiratory rate (RR) reveal its true state [11,32]. As the threat intensity increases, a confined cat may primarily react through an inhibitory pathway, leading to changes in body posture and behavior. This phenomenon aligns with the concept of control and predictability, suggesting that animals are more vulnerable to stimuli initially due to the novelty and lack of familiarity with their surroundings. Although these effects were not statistically significant, the observed increase in response was more pronounced in cats exposed to higher noise levels (G2, G3), despite noise levels being similar at T1, T2, and T3 for all groups. This association highlights the significance of noise level as an environmental factor influencing stress response system (SRS) parameters in hospitalized cats. Based on our results, it seems there may be grounds to consider suggesting changes to FSV scoring. The study observed varying FSV levels among cats exposed to different environmental noise levels. Cats in the control group exhibited considerably lower FSV levels compared to those in other groups, indicating differing stress responses based on environmental conditions. The FSV values tended to increase in response to higher noise levels, suggesting a correlation between FSV and environmental stressors factors. Cats in the different groups considered exhibited varying behavioral responses, with some showing tendencies towards freezing behavior or relaxation, which could influence the FSV scores. Thus, this may warrant adjustments to better reflect the cats’ stress levels accurately. While FSV showed a strong positive correlation with respiratory rate (RR), no significant correlation was observed between FSV and plasmatic cortisol levels. This suggests that FSV may capture certain aspects of stress not reflected in cortisol levels, highlighting its potential value in stress assessment. In conclusion, considering the observed variations in FSV levels in response to environmental factors and its association with other stress parameters, it may be worth exploring potential modifications to FSV scoring methods. However, any changes should be carefully evaluated to ensure reliability, validity, and practicality in clinical or research settings. Further investigation and collaboration with experts in the field may help refine FSV scoring and enhance its utility in assessing stress in cats. It is important to note that pain triggers an increase in cortisol levels, considering that it is a stress hormone. As the anesthesia diminishes, the animal becomes more aware of any discomfort or pain resulting from the surgical procedure, leading to a stress response characterized by elevated plasma cortisol levels. However, in our study, analgesic drugs such as non-steroidal anti-inflammatory drugs and opioids were used pre-operatively to ensure the comfort of the patients in their post-operative period. According to our results, although cortisolemia tended to be higher in G1 cats compared to CG cats at T0 and T3, these differences were not statistically significant [18,33,34]. This supports the notion of habituation of the cortisol response over time as patients are repeatedly exposed to physical stressor parameters [2,25,26,27,28,35,36,37]. The Spearman analysis revealed no association or weak correlation between the final stress value (FSV) and plasmatic cortisol level ([Cort]p), as well as between respiratory rate (RR) and [Cort]p. This lack of correlation may be attributed to variations in stress intensity. McCobb et al. [38] reported no correlation between the Cat Stress Score and cortisol–creatinine ratio, suggesting that stress assessment scores may not always align with physiological markers. In contrast, a strong positive correlation was observed between FSV and RR. This suggests a robust connection between these variables, indicating that changes in stress level are associated with alterations in respiratory rate. Considering that isoflurane is a volatile anesthetic, it is expected that as its effects wear off, the respiratory system may respond by increasing the respiratory rate, potentially as a compensatory mechanism to help clear the anesthesia from the body and restore the normal breathing pattern. The amygdala, insular cortex, and thalamic relay nuclei are components of the limbic circuitry involved in regulating breathing, a crucial physiological function for maintaining acid-base balance in the body [39,40]. Disruptions in these brain regions can lead to alterations in respiratory rate and pattern [41,42,43]. Respiratory rate is influenced by various factors and reflects the activity of multiple homeostatic mechanisms. Elevated stress levels typically lead to increased respiratory rate, often resulting in tachypnea, defined as a respiratory rate higher than normal. Tachypnea can be a response to perceived threats experienced by the animal [44]. Therefore, respiratory rate serves as a valuable tool for assessing animal stress levels, given its sensitivity to stress-induced changes [45,46], such as, identifying freeze behaviors in this species. Moreover, it underscored the impact of stressful events occurring during the preceding experimental period [11,32]. The discussion on whether pain, discomfort, and stress could induce different or overlapping responses in respiratory rate (RR), final stress value (FSV), or cortisol levels in cats is an interesting and difficult topic. Distinguishing between these factors in a clinical or research context can be challenging [47,48,49]. Further investigation utilizing a combination of physiological, behavioral, and hormonal measures is warranted to better understand the distinct and overlapping responses of cats to different stressors and their implications for feline health and welfare.

## 5. Conclusions

The study findings revealed that higher noise levels were linked to elevated values of final stress value (FSV), respiratory rate (RR), and plasmatic cortisol levels ([Cort]p). Additionally, FSV, RR, and [Cort]p values tended to increase with longer exposure to noise. The excellent correlation between FSV and RR suggests that RR can serve as a valuable indicator for evaluating stress levels in cats. Factors such as the presence of other people and animals in the ward should also be considered as stress stimuli. Consequently, isolating cats in areas with low noise levels and restricted access appears to be an effective strategy for reducing stress during hospitalization. These findings provide scientific support for many existing recommendations aimed at improving the welfare of cats in hospital environments, including minimizing noise levels.

## Figures and Tables

**Figure 1 vetsci-11-00173-f001:**
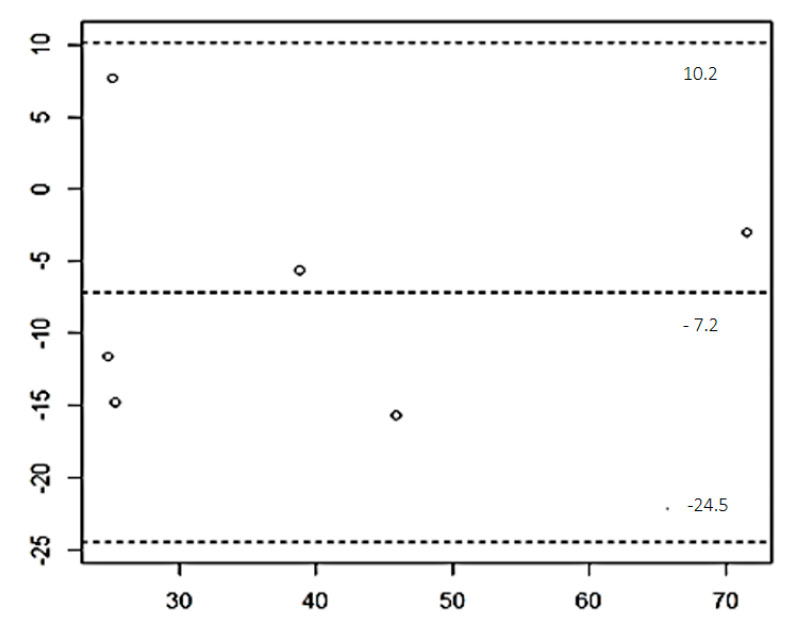
Bland–Altman plot of the results obtained with the laboratory technique ELISA and the commercial kit EIA-1887.

**Table 1 vetsci-11-00173-t001:** The ethogram adapted for this study. In comparison with the original ethogram, the following groups of behavioral categories have been discarded: head, stomach, legs, and vocal.

**Activity**	Sleeping or resting; without body tension	0	**Observations** [18] Sleeping—lying; minimal head or leg movement, and the cat is not easily disturbed. Investigating—cat shows attention toward a specific stimulus. Threatening—cat directs fear aggression behaviors toward without making any physical contact with it. Lying—cat’s body is on the ground in a horizontal position. Sitting—cat is in an upright position, with the hind legs flexed and resting on the ground, while front legs are extended and straight. Standing—cat is in an upright position and immobile, with all four paws on the ground and legs extended, supporting the body. Crouching—cat positions the body close to the ground, whereby all four legs are bent and the belly is touching the ground or slightly raised. Arching back—cat curves back upwards and stands rigidly. Ears erecting—pointed upwards vertically. Ears flattening—close to the head; tend to be level with the top of the head. Tail waving—a slow and gentle waving of the tail from side to side. Tail twitching—rapid flick of the tail in either a side to side or up to down motion.
	Alert or investigating; minimal body tension	1	
	Subtle signs of stress; restless; less responsive to stress stimuli (noise level); mild body tension	2	
	Motionless and/or isolated; decreased responsiveness to stress stimuli (noise level); moderate body tension	3	
	Threatening when approached; high body tension	4	
**Body**	Lying out on side	0	
	Lying ventrally or sitting	1	
	Standing		
	horizontal dorsum	2	
	anterior portion higher than the rear	2.5	
	Crouch	3	
	Arch back	4	
**Eyes**	Closed or half opened	0	
	Opened	1	
	Widely opened	2	
**Pupils**	Normal	0	
	Dilated	1	
**Whiskers**	Relaxed (lateral)	0	
	Forward	1	
	Back (near the face)	2	
**Ears**	Relaxed (half-back)	0	
	Erect	1	
	Partially flattened	2	
	Flattened	3	
**Tail**	Extended or loosely wrapped	0	
	Tense		
	tail waving	1	
	close to the body	1.5	
	Twitch	2	

**Table 2 vetsci-11-00173-t002:** Median and standard deviation, and *t*-test for paired sample applied to both techniques used to measure [Cort]p—Commercial ELISA kit EIA-1887, and the chemiluminescent ELISA system.

Type of Technique	N	Median	SD	*t*-Test for Paired Samples (*p*-Value)
Commercial ELISA kit EIA-1887	6	45.15	7.78	0.68
Chemiluminescent ELISA system	6	35.0	7.63	0.17

Sample (*N*); standard-deviation (SD).

**Table 3 vetsci-11-00173-t003:** Results of Kruskal–Wallis test for the FSV, respiratory rate (RR). and cortisol plasmatic concentration ([Cort]p) comparison between groups.

Groups		*N*	Kruskal–Wallis Test						ANOVA One-Way		
			FSV			RR			[Cort]p		
			Median	IQR	*p*-Value	Median	IQR	*p*-Value	Mean	SD	*p*-Value
**T1**	CG	9	3.0	1.0	<0.001 *	22.0	5.0	<0.001 *	18.63	0.73	<0.01 *
	G1	8	3.5	1.38		23.5	10.25		30.81	0.86	
	G2	8	9.5	1.13		35	13		32.93	2.09	
	G3	8	12.5	0.38		43.0	7.5		37.60	0.75	
**T2**	CG	9	2.0	0.0	<0.001 *	20.0	3.0	<0.001 *	-	-	-
	G1	8	3.0	1.25		28.0	14.5		-	-	
	G2	8	8.5	1.38		41.0	10.0		-	-	
	G3	8	13.5	1.75		45	8.5		-	-	
**T3**	CG	9	1.0	1.0	<0.001 *	20.0	2.0	<0.001 *	27.41	2.50	0.03 *
	G1	8	3.0	1.0		30.0	9.0		38.53	8.94	
	G2	8	9.5	1.5		35.0	12.0		42.37	1.29	
	G3	8	13.0	2.5		57.0	7.5		46.03	1.92	

Sample (*N*); CG (cats placed in a room isolated from other animals); G1 (cats placed in a room where noise intensity was less than 60 dB); G2 (cats placed in a room with noise intensity between 60 and 85 dB); and G3 (cats placed in a room where noise intensity was higher than 85 dB); IQR (interquartile range). * *p*-value statistically significant.

**Table 4 vetsci-11-00173-t004:** Results of the non-parametric post-hoc tests of Dwass–Steel–Critchlow–Fligner (DSCF) and Bonferroni carried out to compare the values of the variables FSV (final stress value), respiratory rate (RR) and the plasmatic cortisol concentration ([Cort]p) obtained for specific pairs of group at specific time points. [Cort]p. Spearman’s correlation between the stress markers used in the study.

Time Points Post-Surgery	Comparisons of Group Pairs	Post-Hoc Dwass–Steel–Critchlow–Fligner (DSCF) Test				Post-Hoc Bonferroni Test	Spearman’s Coefficient Test	
		FSV		RR		[Cort]p	FSV, RR, [Cort]p	
		|D|	Cut-Off	|D|	Cut-Off	*p*-Value	rho	
**T1**	CGG1	2.33	>0.37	3.82	>0.02	< 0.01 *	-	
	CGG2	5.26	>0.0	4.46	>0.0	0.18	-	
	CGG3	4.57	>0.0	4.51	>0.0	0.24	-	
	G1G2	5.07	>0.79	0.63 *	>0.98 *	0.13	-	
	G1G3	4.43	>0.11	3.30	>0.08	0.20	-	
	G2G3	4.58	>0.18	2.62	>0.26	0.05	-	
**T2**	CGG1	3.06	>0.12	1.73	>0.64	-		
	CGG2	5.29	>0.0	4.63	>0.0	-		
	CGG3	4.79	>0.0	4.72	>0.0	-		
	G1G2	4.91	>0.70	3.60	>0.04	-		
	G1G3	4.46	>0.25	3.98	>0.01	-		
	G2G3	4.77	>0.17	0.69 *	>0.97 *	-		
**T3**	CGG1	2.96	0.15	2.82	0.19	0.03 *	FSV—[Cort]p	0.09
	CGG2	5.14	>0.0	4.82	>0.0	-		
	CGG3	4.59	>0.0	4.52	>0.0	-	RR—[Cort]p	0.20
	G1G2	5.12	0.89	2.82	0.20	0.20		
	G1G3	4.55	0.66	4.26	0.02	0.50	FSV—RR	0.91
	G2G3	4.55	0.06	3.67	0.04	0.13		
**T1–T3**	CG	-	-	-	-	0.18		
	G1	-	-	-	-	0.63		
	G2	-	-	-	-	0.31		
	G3	-	-	-	-	0.61		

Sample (*N*); The different time points post-surgery: T1 (1 h post-surgery), T2 (2 h post-surgery), and T3 (3 h post-surgery). CG (cats placed in a room isolated from other animals); G1 (cats placed in a room where noise intensity was less than 60 dB); G2 (cats placed in a room with noise intensity between 60 and 85 dB); and G3 (cats placed in a room where noise intensity was higher than 85 dB); FSV (final stress value); RR (respiratory rate); [Cort]p (plasmatic cortisol concentration). * Statistically significant results because |D| > cut-off.

## Data Availability

The raw data supporting the conclusions of this article will be made available by the authors on request.

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
