# Peer review of "The Influence of Noise Level on the Stress Response of Hospitalized Cats"

_vetsci, 2024, doi:10.3390/vetsci11040173_

Round 1

Reviewer 1 Report

Comments and Suggestions for Authors

This is a well written paper, congratulations

Most of my commentary is associated with more detail in the methods

Please describe

1) where the study took place

2) the experience and number of different observers who characterised stress levels in cats and whether such observers worked across all four noise zones

3) How long were the FSV observations conducted for at each time period?

4) How was RR calculated and what are the units? 

5) The origin and nature of the noises in the different treatments? Readers should be aware if they were consistent background noises or intermittent 'bangs'

6) whether 'complications' or ideosynchracies were recorded for the different operations and whether these were controlled for or could have influenced outcomes

In the discussion it would be good to discuss whether pain, discomfort and stress could induce different or overlapping responses in RR, FSV or cortisol levels.

Do you suggest any changes to the FSV scoring as a result of your studies?

Given your findings that RR provides a useful index of stress it would be valuable to describe how RR could be assessed by vets and cat carers and what levels indicate different levels of stress.

Line 105  Stanton et al ref should be included in References

Author Response

Dear Reviewer,

I hope this message finds you well. Thank you for taking the time to review our manuscript titled "THE INFLUENCE OF NOISE LEVEL ON THE STRESS RESPONSE OF HOSPITALIZED CATS" which we have submitted for publication in the vetsci journal.

We appreciate your valuable feedback, and we have carefully considered all of your comments. In response, we have made revisions to the manuscript to address each point raised. Below, you will find our detailed responses to your comments, organized to correspond with the changes made in the original version of the manuscript.

Thank you once again for your time and thoughtful input.

Best regards,

L.Miguel Carreira

Reviewer 2 Report

Comments and Suggestions for Authors

1、            The author simulated the impact of noise on cats after surgery by observing the effects of different noise decibel levels on cats at one hour, two hours, and three hours after the operation, respectively. What are the scientific bases for the decibel levels and durations you have chosen

2、            Why didn't you set up a group of cats that underwent surgery and were exposed to the same decibel level as the control group? Please indicate the dosage of isoflurane in the materials and methods section. As I understand, isoflurane is an anesthetic that induces rapid recovery but has a slight respiratory depressant effect. Is it possible that the increase in respiratory rate at the third hour was due to the waning of the anesthetic effect? Please explain whether it is possible that the decrease in the effect of isoflurane at the third hour postoperatively led to an increase in cortisol levels due to pain caused by the subsiding anesthesia in the animal's body?

3、            I believe that a mere elevation in cortisol levels cannot directly prove the activation of the hypothalamus-adrenal-adrenal cortex axis. Further experiments are needed to provide evidence for this.

4、            I suggest that you might want to try applying this method to cats that have undergone testicular removal. We need to further understand the impact of this method on male cats. Additionally, increasing the number of animals would be beneficial for reducing errors and improving the quality of the work.

Author Response

(The authors gave the same response as above.)

Reviewer 3 Report

Comments and Suggestions for Authors

A very ingenious study on cats' susceptibility to noise-induced stress while recovering from surgery.
I have a few comments and doubts:
- the methodology does not clearly describe who assessed the behavior before the procedure (5 and 10 minutes after being placed in the cage) and after the procedure (immediately, after 2 and 3 hours) - were they the same people? what competences did they have?
- why wasn't the heart rate measured?
- why was body temperature not measured?
- how was blood collected to measure plasma cortisol concentration? Were the cats anesthetized again or were they forcibly immobilized, how long did the blood collection take, how many people participated in the collection?
- Were painkillers used? If so, what and in what doses? After all, pain after surgery causes stress.
In my opinion, this is a much stronger stressor than noise!
- Were any protectors used after waking up to prevent licking the wound? (collar around the neck or caftan for the torso).
- it is not described what technique was used to castrate females (laparoscopy? classic surgery?)
  - was the procedure time taken into account (ovariohysterectomy may last from several to several dozen minutes)
- what was the source of the noise and how was it measured?
  There is no information in the "Material and methods" chapter about two methods for determining cortisol, only in the "Results" chapter there is a description of the ELISA kit and the chemiluminescent ELISA system presented in the table. 2.

Author Response

(The authors gave the same response as above.)

Reviewer 4 Report

Comments and Suggestions for Authors

25 and through out plasma not plasmatic

72 remove first phrase "Referenced...cats"

82 these were not pet cats?   who were the tutors? 

Methods Were noise levels actually measured? This is an important point. Decibel level mentioned in abstract, but not methods. What instrument was used to measure noise?

What was the source of noise- other cats? machinery?  people talking?

Cg were these cats in a room alone whereas the other groups  G2-G4 were in rooms with other cats?

What is {D} cutoff?

Table 1

corporal should be body 

Really don't understand tail level with top of head?

Even native English speakers have trouble with this, but the verb to lie is lie present tense; lay past tense, have lain past perfect 

lay means to put something down. I lay the phone down. I laid the phone on the table yesterday, Use lying. Lying on side lying ventrally ( on sternum). Sitting 

Table 3 omit considered groups 

86 avoidance of not to

88 how many cats were excluded?

110 between observers?

134 were cats in Other groups not isolated?  When isolated was each cat in a room alone?

152 laboratory not laboratorial

163 agreement not clinical significance

165 validated instead of certified

185 and surgery!

191 stored as 

G1, G2, and G3 tended to increase at T1, T2, and T3 in response to the increasing noise level. This is confusing The response increase was higher in cats exposed to more noise (G2 G3) not because noise was higher at T1 T2 and T3 for all groups 

218 this implies the noise levels increased with tie not room. Is that true?

221Regarding cortisol levels, although cortisolemia tended to be higher Rewrite Although cortisol levels tended...

need a reference to the Final Stress Value ( Turner?)

Comments on the Quality of English Language

see corrections in comments to author

Author Response

(The authors gave the same response as above.)
